# Adipose Tissue-Derived Stem Cells: The Biologic Basis and Future Directions for Tissue Engineering

**DOI:** 10.3390/ma13143210

**Published:** 2020-07-18

**Authors:** Diana Aparecida Dias Câmara, Jamil Awad Shibli, Eduardo Alexandre Müller, Paulo Luiz De-Sá-Junior, Allan Saj Porcacchia, Alberto Blay, Nelson Foresto Lizier

**Affiliations:** 1Nicell-Pesquisa e Desenvolvimento Científico LTDA, São Paulo 04006-000, Brazil; nlizier@gmail.com; 2M3 Health Ind. Com. de Prod. Med. Odont. e Correlatos S.A., Jundiaí 13212-213, Brazil; alberto.blay@plenum.bio; 3Department of Periodontology and Oral Implantology, Dental Research Division, University of Guarulhos, Guarulhos 07040-170, Brazil; dre27bq@hotmail.com; 4Villa Lobos Campus, University Mogi das Cruzes (UMC), São Paulo 08780-911, Brazil; paulsaj2001@yahoo.com.br; 5Department of Psychobiology, Federal University of São Paulo, São Paulo 04021-001, Brazil; allansaj.7@gmail.com

**Keywords:** adipose-derived stem cells, heterogeneity, tissue engineering

## Abstract

Mesenchymal stem cells (MSCs) have been isolated from a variety of tissues using different methods. Active research have confirmed that the most accessible site to collect them is the adipose tissue; which has a significantly higher concentration of MSCs. Moreover; harvesting from adipose tissue is less invasive; there are no ethical limitations and a lower risk of severe complications. These adipose-derived stem cells (ASCs) are also able to increase at higher rates and showing telomerase activity, which acts by maintaining the DNA stability during cell divisions. Adipose-derived stem cells secret molecules that show important function in other cells vitality and mechanisms associated with the immune system, central nervous system, the heart and several muscles. They release cytokines involved in pro/anti-inflammatory, angiogenic and hematopoietic processes. Adipose-derived stem cells also have immunosuppressive properties and have been reported to be “immune privileged” since they show negative or low expression of human leukocyte antigens. Translational medicine and basic research projects can take advantage of bioprinting. This technology allows precise control for both scaffolds and cells. The properties of cell adhesion, migration, maturation, proliferation, mimicry of cell microenvironment, and differentiation should be promoted by the printed biomaterial used in tissue engineering. Self-renewal and potency are presented by MSCs, which implies in an open-source for 3D bioprinting and regenerative medicine. Considering these features and necessities, ASCs can be applied in the designing of tissue engineering products. Understanding the heterogeneity of ASCs and optimizing their properties can contribute to making the best therapeutic use of these cells and opening new paths to make tissue engineering even more useful.

## 1. Introduction

The defining characteristics of mesenchymal stem cells (MSCs) are their capacity to self-renew and their multipotency to differentiate into more than one cell type and remain in this state for long periods [1]. Furthermore, MSCs produce growth factors and cytokines that are involved in immunomodulation and regeneration. This immunomodulatory capacity of MSCs enables them to be used in cell therapies, especially in autoimmune diseases, host grafting and organ transplantation [2]. In addition, tissue-derived stem cells have a degree of plasticity, depending on their type. This manifests in the differentiation phenotypic potential that goes beyond the cell phenotype of their original tissue [3].

Mesenchymal stem cells have been isolated from several different tissues [4,5] using a variety of different methods, although the most accessible site is adipose tissue [6]. Adipose tissue has a significantly higher concentration of MSCs than bone marrow (1% versus > 0.01%) and other sources, including the dermis, the umbilical cord, dental pulp and the placenta [7]. Moreover, harvesting from adipose tissue is less invasive when compared to the bone marrow, resulting in less risk of severe complications and no ethical limitations [6]. It was established that the positive expression of surface markers CD13, CD29, CD44, CD73, CD90 and CD105, and negative or low production of HLA-DR characterizes MSCs [8,9]. Adipose tissue (AT)-derived MSCs tend to be more heterogeneous [10] and exhibit immunomodulatory characteristics, in addition to their differentiation ability similar to bone marrow-derived MSCs (BM-MSCs) [11].

To prevent the incorrect use of different and diverse terminology, the International Fat Applied Technology Society adopted the term “adipose-derived stem cells” (ASCs) to identify the isolated, plastic-adherent and multipotent cell population obtained from this site (Figure 1) [12].

Adipose tissue is a complex connective tissue that originated from the mesodermal: an energy homeostasis regulator, which exhibits morphologic, functional and regulatory heterogeneity [13]. Several types of cells compose AT, including preadipocytes, mature adipocytes, vascular smooth muscle cells, fibroblasts, resident monocytes, endothelial cells, macrophages and lymphocytes [14]. The immune, endocrine, reproductive, and hematopoietic systems are influenced by AT, acting in the inflammatory response and many other functions [15].

Understanding the heterogeneity of ASCs and how to optimize their properties, can contribute to making the best clinical use of them and lead to more effective tissue engineering.

## 2. Sources of Heterogeneity

Adipose-derived stem cells are heterogeneous and exhibit various features, such as proliferation capacity, differentiation potential, expression of specific surface immunophenotypes and a secretive profile. These variations exist due to multiple features such as the donor age, gender, the body mass index, the patient’s clinical condition, the isolation procedure, in which body site and depth of each adipose depot and the sample were withdrawn. Additionally, the cell culture methodology, including the type of medium and the culture surface, are also factors that may influence and therefore affect the main phenotypic characteristics of ASCs [16] (Figure 2).

This heterogeneity probably occurs due to the differential behavior of cells from each depot source. Plenty of studies suggested that ASC exhibit distinct depot-specific gene expression profiles among adipose tissue depots, apart from differences in adipogenic and in immunomodulatory potentials [17,18]. The key to the establishment of morphofunctional heterogeneity between white adipose tissue (WAT) and brown adipose tissue (BAT) depots is based on the development of depot-specific responses to metabolic challenges [19].

In humans, each of the two significant stores of WAT, the visceral and the subcutaneous, show different structures, gene expression, cell content, secretion profiles and responsiveness to neuroendocrine stimuli. The hypodermis, formed by the subcutaneous WAT, is distributed along the body surface and form establishes distinct stores in the cranial, facial, gluteal, femoral and abdominal areas [20,21].

White adipose tissue is highly plastic and can undergo extensive remodeling in response to metabolic changes, nutritional and pharmacological challenges, thus influencing the number of ASCs and their microenvironment [22].

Brown adipose tissue is found in the beginning of life and during growth, until adulthood, it is transformed into WAT [23]. The most common location for BAT, which was detected in adults by tomography scan, was the supraclavicular depot, in a distinct fascial plane in the ventral neck, superficial and lateral to the sternocleidomastoid muscles. Moreover, pieces of evidence reported that BAT is still functional in the regions, and can play a regulatory role in energy metabolism [24,25].

## 3. Cellular Heterogeneity

Adipose-derived stem cells are a potent cell population. They are influenced by environment and genetic composition and display differences to their niche in the body, as well as proliferation capacity and stemness [26]. Furthermore, the same AT isolation site may contain several types of ASCs, and as it is a rich cell niche, each cell may lead to a different possible therapeutic application [27].

The exact site of ASCs into adipose tissue has not been well defined, and therefore, the composition of the adipose tissue depends on each individual group. One of the reasons is the high tissue vascularization and the presence of the various cell types [27]. It is known, nonetheless, that AT includes endothelial cells, immune cells, pericytes, fibroblasts, vascular cells, preadipocytes ASCs and hematopoietic stem cells. Stem Cell and progenitor cells resolve about 3% of all cell populations [28]. The process of adipogenesis and angiogenesis can be closely related, moreover pericytes exhibit multipotency similar to that of stem cells [29,30]. Importantly, these stromal and immune cell types play critical roles in the establishment and maintenance of parenchymal cell function. The composition of stromal cells varies across fat depots, reflecting tissue specialization and differences in energy storage, vascularization, innervation and metabolism [31,32].

Owing to its essential role in metabolism, there is considerable interest in better defining cellular subtypes involved in AT homeostasis and the mechanisms that regulate in vivo adipogenesis, plasticity and inflammatory processes for targeted therapeutic strategies to treat metabolic and autoimmune disease [20].

## 4. Adipose-Derived Stem Cell

Correlated with BM-MSCs, ASCs exhibit a greater proliferative capacity and present telomerase activity. Even though this activity is lower than the one observed in tumor cell lines, it confirms the ASCs self-renewal and proliferation ability [33]. Adipose-derived stem cells advocate tissue regeneration and repair by secreting growth factors, cytokines, angiogenic factors, adipokines and neurotrophic factors that stimulate restoration of normal tissue function or reduce the damage [26]. Molecules released by ASCs play essential roles in the vitality of other cells and mechanisms associated with central nervous system, immune system, heart and muscles [34]. The cytokine profile of ASCS comprises, pro/anti-inflammatory, angiogenic and hematopoietic factors, being interleukins (IL-6, IL-7, IL-10, IL-11), vascular endothelial growth factor (VEGF), basic fibroblastic growth factor (bFGF), tumor necrosis factor-alpha (TNF-α), granulocyte colony-stimulating factor (G-CSF) and macrophage colony-stimulating factor (M-CSF) [35]. The immunosuppressive properties of ASCs can result from the release of Indoleamine-2,3-dioxygenase and prostaglandin E2 [36,37].

Through the secretion of brain-derived neurotrophic factor, glial-derived neurotrophic factor, nerve growth factor and insulin-like growth factor (IGF) [34], ASCs show neuroprotective action and promote the regeneration of central nervous system tissues.

Adipose-derived stem cells are also immune-privileged due to the negative or low expression of human leukocyte antigen (HLA) expression and inhibition of proliferation of activated allogeneic lymphocytes [38,39]. These cells exert protector effects against organ rejection and avoid graft versus host disease after allogeneic transplantation.

The immunophenotype of ASCs can express other essential factors involved in stemness, self-renewal and differentiation potentials, such as CD146 and CD271. They are also related to enhanced capacity for healing bone defects or cartilage or to differential paracrine wound healing activity [40,41].

Li and collaborators isolated a CD146^+^ subpopulation from ASCs and tested for cartilage regeneration [40]. Cartilage lesions typically result in inflammation and represent an important test in cartilage repair [42]. The inflammation-modulating property in the animal model reported better results during the early stage of intra-articular injections of CD146^+^ ASCs.

In another study, Kohli and collaborators showed that derived CD271^+^ ASCs were demonstrated to be macroscopically superior for promoting cartilage repair of osteochondral tissue damage when compared with defects received transplants of plastic adherent ASCs and the control group of scaffold alone [41]. Imperatively, in animal cell transplantation groups, there was slight evidence of mature hyaline cartilage or new bone tissue.

These studies emphasized the importance of discovering the functional features of specific subtypes of ASCs, to be used as a new path for cell-based tissue engineering research.

## 5. Biomaterials

Several materials with the ability to mimic the ECM have been investigated for scaffold production, potentiating the effects of ASCs. These materials range from decellularized tissue matrices to inorganic ceramics, such as bioceramics with application in hard tissue replacement orthodontic; polymers- natural proteins, polysaccharides with applications in connective and hard tissues, decellularized living tissues/organs and drug delivery; polymers synthetic degradable with application in implants and non-degradable for orthopedic implants; metals for orthopedic and dental application; composites with orthopedic and dental application [43,44,45].

Natural and synthetic biomaterials can be formed, producing bioactive scaffolds that control stem cell differentiation into the desired tissue type. These scaffolds can be made in several forms, which display unique features. Scaffolds must consider three requirements: mechanical (rigidity, modulus of elasticity), physical–chemical and biologic characteristics [46]. The design and construction of scaffolds used to repair damaged tissue must be done in a personalized way to maintain the similarity to the anatomic structure and better to perform the biomechanical functions of the pristine tissue. The 3D scaffold must temporarily resist the external stress and pressures caused during the new tissue formation, retaining biomechanical characteristics similar to those of the adjacent tissue [47].

The biomaterial degradation may be found by biologic, physical, chemical or combined processes affecting the biocompatibility of the scaffold. In cases where a complete degradation is not necessary in the application of a scaffold, for example, semi-permanent or permanent scaffolds could be used. Foreign body reaction, immunological or toxic responses should not emerge after the scaffold implantation, which proves its biocompatibility. It is essential to consider cell attachment, homogeneous distribution, proliferation and cell-to-cell contacts, all cell features, during the design of the scaffold surface. The scaffold microstructure should preserve the porous or fibrous design and allow a high surface-to-volume ratio for cell attachment and tissue formation. Polished materials do not present upper surface energy as demonstrated by nanostrutured surfaces, which results in higher hydrophilicity and, consequently, enhance adhesion of proteins and cell attachment [45]. Complementary, ceramic and metal scaffolds have shown that smaller grain size not only raises the mechanical strength but was found to be favorable in terms of attachment and proliferation of osteogenic cells. Nonetheless, cell behavior is highly associated with the scaffold topography with its mechanical features [46]. Cells seeded in 3D scaffolds, including MSCs, demand to be favored to regain typical in vivo morphology and this is what makes a successful treatment.

## 6. Impact on Tissue Engineering

Three-dimensional (3D) frameworks are superior when compared to two-dimensional (2D) cultures and that exert a significant influence on cellular processes [48] (Figure 3). The 3D structures are multifunctional and mechanically robust. Therefore, they offer the type of substrate appropriate for the growth of MSCs, in addition to replicating the specific physiological environment for the cells during culture [44,45].

As a type of approach to biofabrication, bioprinting hold preferences of high throughput and exact control of both cells and scaffolds. This technical knowledge is not only crucial for translational medicine, but also basic research applications. Bioprinting has been extensively used to build functional tissues such as bone, muscle, cartilage and vasculature [49,50]. This technology can be involved in complex tissue structure fabrication (based on the converted medical images) and as an efficient tool for drug encounter and preclinical testing. The biomaterials and stem cell biology fields have been extensively explored to combine several printing mechanisms into multi-phasic tissue engineering [43].

Biomaterials produced by additive manufacturing and 3D printing technologies used in tissue engineering should also have properties allowing them to differentiate, as well as mimic the cellular microenvironment so that they can transport stem cell regulatory signals in a perfect and near-physiological fashion [51].

Developments in biomaterials support the design and progress of ideas based on biomaterial scaffolds for ex vivo cell expansion. An example would be the combination of those biomaterials with a battery of inductors, which can lead to the differentiation of MSCs into a specific lineage [52].

Biomaterial scaffolds conjugated with growth factors and co-printing cells result in a very effective tissue engineering modality with a consistent supply of biomimetic cues during the long-term regeneration process. The selection of the material and composition of the scaffold are significant hurdles for bioprinting to overcome [53]. The most complex issue is that the printed tissues have demonstrated reduced survivability and therapeutic efficiency [54].

Differently, Adipose-derived stem cells present characteristics of self-renewal and potency, showing an immensurable cell source for 3D bioprinting and regenerative medicine, on the leading edge of the development of new applications for ASCs in designing tissue engineering products.

The scaffold structure positively influences cell cultures in the scaffold, and characteristics such as fiber thickness, pore size, porosity, topography and scaffold stiffness directly impact cell behavior and colonization [55].

Many studies have indicated the osteogenic and chondrogenic differentiation potential of ASCs when cultured in several scaffolds from organic and inorganic sources, depicting that this combination strongly improved the cell differentiation achievement [53,54,55]. The paracrine signals of ASCs recruit progenitor cells from the host tissues and enhance regeneration, accelerating the vascularization of the implanted area, promoting matrix mineralization, improving bone formation and cartilage repair [56,57,58].

An interim 3D construct or scaffold is created to biomimetize the extracellular matrix (ECM) to provide mechanical support. It should guide the healing process and promote the differentiation of progenitor cells. It must also be biocompatible and its microstructure needs to allow enough pore sizes and distribution with fine interconnectivity for nutrient and oxygen transport to cells, thereby leading to proper cell attachment and growth [45,59] (Figure 4). In addition, the network structures of the pores also help in guiding and supporting new tissue formation. 

Medicine becomes more personalized and precise by employing scientific knowledge and advanced scaffolds for its purposes. More specifically, scaffolds can be applied in research projects for healing damaged tissues, synthesize molecules, cell products, cells, and induce factors for regeneration of human tissues.

## 7. Conclusions

Alterations in the culture settings of ASCs—despite the phenotype of cells after isolation—can alter the cell profile to various subpopulations. This can allow considerable differences in the proliferation rate and differentiation of cells available for therapeutic procedures, offering several new venues for tissue engineering and regenerative medicine strategies, especially in precision medicine.

Mesenchymal stem cell administration can be done through different methods. After the in vitro expansion, a practical approach is to combine ASCs with scaffolds, which can be made from natural or synthetic materials. The natural processes of tissue regeneration that exist in the human body can be mimicked and enhanced in the combination of stem cells and scaffolds when designed for therapeutic purposes. In this way, bioengineered scaffolds have the potential to mimic natural signaling and repair processes to produce a microenvironment for adhesion, proliferation, differentiation of stem cells that regenerate the damaged tissues. In addition, stem cells metabolism and differentiations can be controlled and stimulated by adding several molecules, such as peptides, drugs, antibodies, growth factors, mRNAs, genes and other molecules of importance.

## Figures and Tables

**Figure 1 materials-13-03210-f001:**
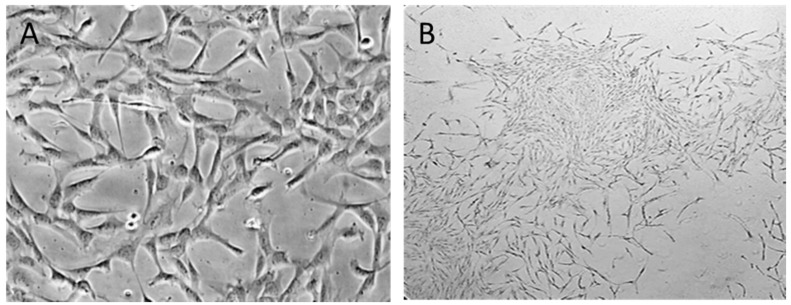
Morphology of ASCs. (**A**) Isolated ASCs are typically expanded in monolayer (2D) culture plastic (original 20× magnification) and (**B**) ASCs colony-forming potential (original 4× magnification).

**Figure 2 materials-13-03210-f002:**
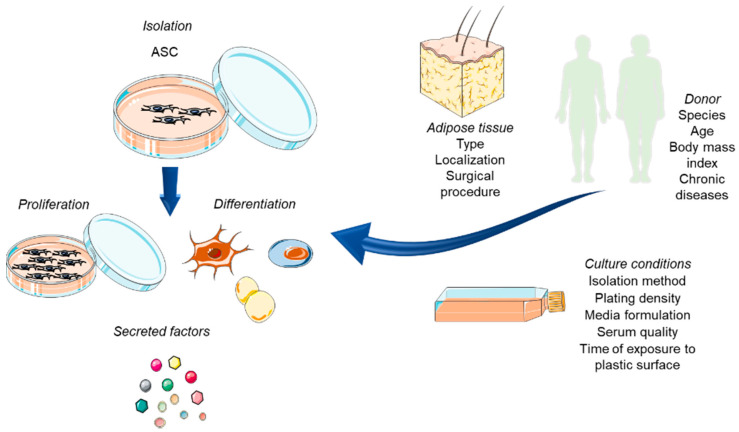
Isolation procedure of ASC. The variety of sources and the diversity of factors influencing ASCs biologic properties (proliferation capacity, differentiation potential, secretion of factors), increasing heterogeneity and leading to the isolation of differentiated subpopulations.

**Figure 3 materials-13-03210-f003:**
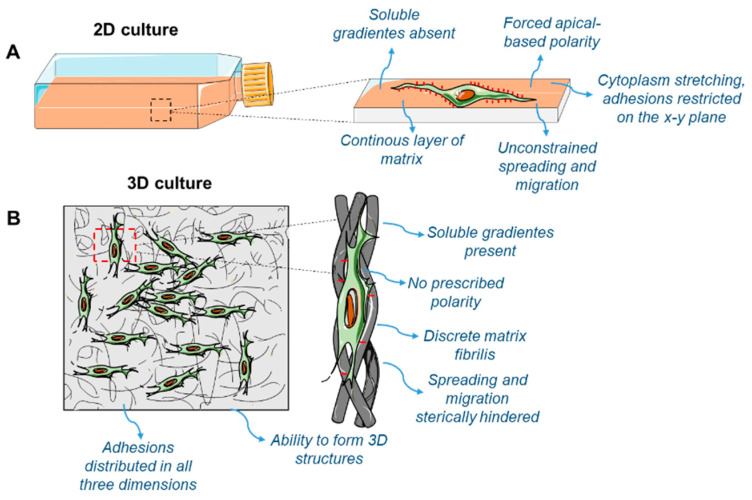
Comparison between 2D-cell culture (**A**) and 3D-cell culture (**B**). Cells cultivated in 2D demonstrate environmental behavior and constraints, whereas when grown in 3D culture systems, they feature increased complexity, to simulate increased faithfulness to the in vivo environment.

**Figure 4 materials-13-03210-f004:**
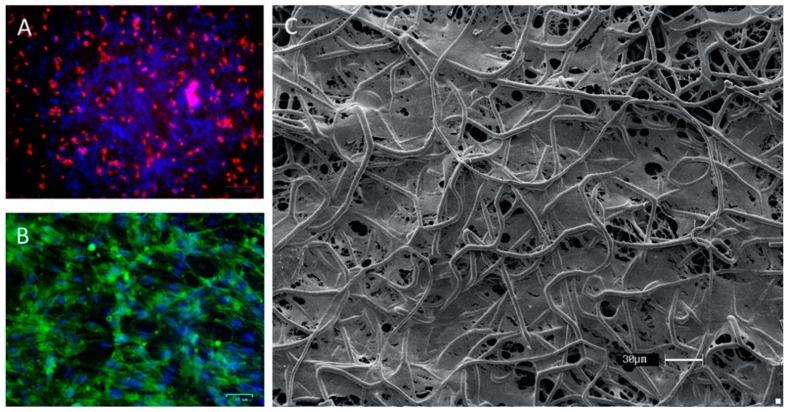
Morphology of culture in the scaffold. (**A**) Fluorescence microscopy of adipose-derived stem cells (ASCs) staining with PKH26 in the first hours of culture in the scaffold; (**B**) fluorescence microscopy of culture of ASC in scaffold staining with phalloidin–FITC; (**C**) Scanning electron microscopy of ASCs on multi-polydioxanone (PDO) scaffold after 14 days of culture.

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
