# Peer review of "Adipose Tissue-Derived Stem Cells: The Biologic Basis and Future Directions for Tissue Engineering"

_materials, 2020, doi:10.3390/ma13143210_

Round 1

Reviewer 1 Report

General Comments

This is a concise review that focuses on ASC as a cell model for use in 3D scaffolds in regenerative medicine.  While the content has been covered by other authors in the past, the authors have managed to highlight recent (2020) publications and incorporated them to address the intersection of ASC with 3D scaffolds.  This is an active area of publication and there have been multiple manuscripts that are relevant to this space which have appeared within the past few months.  While it is impossible to cite everything in the literature, the authors are encouraged to re-check the literature one more time to determine if any recent findings merit incorporation in their judgement (this is a suggestion only).

Specific Comments

Ln 50.  The abbreviation MSC is used for both mesenchymal stem cells and multipotent stromal cells; both definitions should be acknowledged and cited accordingly.  This abbreviation has been the subject of considerable controversy over the past two decades.

Ln 93.   The authors include question marks (?) after several terms but it is unclear what the purpose of this character is meant to be. Please clarify or remove.

Ln 90.  The authors entitled the section “Sources of Homogeneity”; however, the entire section focuses on sources of Heterogeneity.  Revising the subtitle to reflect this content is recommended.

Ln 170.  The term “ASC” is underlined and spelled “ACS”.  Please correct.

Ln 175-177. The authors are referencing IGF-1 for the first time without any evidence presented of its production by ASC.  Additional information should be provided to the reader to present findings in the context of the ASC before introducing concepts relating to cardiomyocytes.

Ln 287, Conclusion.  The authors end the paper somewhat abruptly and focus exclusively on the ASC per se rather than its utility in the context of 3D printed matrices.  It is recommended that the authors include text identifying how 3D matrices interface with ASC and what future research directions they foresee based on the data they have presented to the reader.

Ln 443. Reference 56 is incomplete (no names of authors, abbreviations only)

Author Response

Comments of Reviewer 1

General Comments

This is a concise review that focuses on ASC as a cell model for use in 3D scaffolds in regenerative medicine. While the content has been covered by other authors in the past, the authors have managed to highlight recent (2020) publications and incorporated them to address the intersection of ASC with 3D scaffolds. This is an active area of publication and there have been multiple manuscripts that are relevant to this space which have appeared within the past few months. While it is impossible to cite everything in the literature, the authors are encouraged to re-check the literature one more time to determine if any recent findings merit incorporation in their judgement (this is a suggestion only).

Thank you very much for the comments. ASC and 3D scaffolds are a “hot topic” in the field. We agree with the reviewers comment and the revised manuscript present a pretty new reference in the Introduction section as follow:

P3L58:”… ability to exhibit a phenotypic potential that goes beyond the differentiated cell phenotype of their original tissue [3].”

#3 - Hutchings G, Janowicz K, Moncrieff L, et al. The Proliferation and Differentiation of Adipose-Derived Stem Cells in Neovascularization and Angiogenesis. Int J Mol Sci. 2020;21(11):E3790. Published 2020 May 27. doi:10.3390/ijms21113790

Specific Comments

Ln 50. The abbreviation MSC is used for both mesenchymal stem cells and multipotent stromal cells; both definitions should be acknowledged and cited accordingly. This abbreviation has been the subject of considerable controversy over the past two decades.

Thanks for the comment. We will keep the sentence as set: The defining characteristics of Mesenchymal stem cells (MSCs) are their ability to self-renew and their multipotency, or ability to differentiate into more than one cell type, and remain in this state for long periods (both in vitro and in vivo) [1].

Ln 93. The authors include question marks (?) after several terms but it is unclear what the purpose of this character is meant to be. Please clarify or remove.

The revised version was corrected accordingly.

Ln 90. The authors entitled the section “Sources of Homogeneity”; however, the entire section focuses on sources of Heterogeneity. Revising the subtitle to reflect this content is recommended.

Well pointed. We change the title’s section as suggested: Sources of Heterogeneity

Ln 170. The term “ASC” is underlined and spelled “ACS”. Please correct.

Thanks for the comment. We did the changes as requested.

Ln 175-177. The authors are referencing IGF-1 for the first time without any evidence presented of its production by ASC. Additional information should be provided to the reader to present findings in the context of the ASC before introducing concepts relating to cardiomyocytes.

The sentence was revised as follow: The cytokine profile of ASCs comprises, pro/anti-inflammatory, angiogenic, and hematopoietic factors, such as interleukins (IL-6, IL-7, IL-10, IL-11), vascular endothelial growth factor (VEGF), basic fibroblastic growth factor (bFGF), tumor necrosis factor-alpha (TNF-α), granulocyte colony-stimulating factor (G-CSF), and macrophage colony stimulating factor (M-CSF) [35]. The immunosuppressive properties of ASCs also result from the production of prostaglandin E2 and Indoleamine-2,3-dioxygenase [36,37].

Ln 287, Conclusion. The authors end the paper somewhat abruptly and focus exclusively on the ASC per se rather than its utility in the context of 3D printed matrices. It is recommended that the authors include text identifying how 3D matrices interface with ASC and what future research directions they foresee based on the data they have presented to the reader.

Thanks for the comment. We agree with the reviewer suggestion and the conclusion was revised as follow:

Conclusion

Changes in the culture conditions of ASCs, despite the phenotypic closeness of cells after isolation, can shift the cell profile to several subpopulations, consequently leading to considerable differences in the proliferation and differentiation of cells available for therapeutic procedures, opening multiple novel avenues for tissue engineering and regenerative medicine strategies. Thus, they are potentially further the development of precision medicine.

Administration of mesenchymal stem cells can be done through different methods. After the in vitro expansion, an effective approach is to combine adipose-derived stem cells with scaffolds, which can be made from natural or synthetic materials. The natural processes of tissue regeneration that exists in human body can be mimicked e enhanced in the combination of stem cells and scaffolds when designed to therapeutic purposes. In this way, bioengineered scaffolds have the potential to mimic natural signaling and repair processes to generate a microenvironment for adhesion, proliferation, differentiation of stem cells that regenerate the damaged tissues. In addition, stem cells metabolism and differentiations can be stimulated and controlled by adding a variety of factors, such as peptides, drugs, antibodies, growth factors, genes, mRNAs and other molecules of importance.

Ln 443. Reference 56 is incomplete (no names of authors, abbreviations only)

The reference was revised accordingly: [56] Sheehy EJ, Kelly DJ, O’Brien FJ. Biomaterial-based endochondral bone regeneration: a shift from traditional tissue engineering paradigms to developmentally inspired strategies. Mater Today Bio 2019;3:100009. https://doi.org/10.1016/j.mtbio.2019.100009.

Reviewer 2 Report

In this discursive review of the literature some characteristics of the Adipose-derived stem cells are illustrated, such as heterogeneity, the possible use in association with different biomaterials and the possible application with 3D printing.

The theme that the manuscript deals with could be interesting for the reader and presents innovative aspects, especially with regards to 3D bioprinting.

I encourage the publication of the manuscript as it is.

There are only minor typographical errors in the manuscript

Originality/Novelty: The manuscript proposes, through a discursive review of the literature, an original point of view especially on the theme of the application of Adipose-derived stem cells to 3D bioprinting.

Significance: The discussion of the topic is conducted with scientific consistency and rigor through a discursive review of the literature.

Quality of Presentation: The manuscript is written in an appropriate way and the reference articles are cited correctly and appropriately.

Scientific Soundness: Being a discursive review of the literature, the manuscript cannot be judged according to the parameters that are used in the case of an experimental scientific study. Despite this, the logical thread used for the discussion of the topic appears robust since the results of the scientific articles have been reported correctly and in logical sequence.

Interest to the Readers: In my opinion, the conclusions are very interesting for the readership of the Journal. in fact, in my humble opinion, the manuscript identifies an important and innovative direction for scientific research, namely the use of Adipose-derived stem cells in association with 3D printers for the construction of biological and implantable scaffolds.

Overall Merit: There is an overall benefit to publishing this work.

English Level: The English language is appropriate and clearly understandable.   

Author Response

We appreciate your review, notes, and valuation. We are also glad for the praise you exposed for our manuscript. We looked for solving the typographical errors you pointed, which increases the quality of the article. Best regards and be safe.

General Comments

In this discursive review of the literature some characteristics of the Adipose-derived stem cells are illustrated, such as heterogeneity, the possible use in association with different biomaterials and the possible application with 3D printing.

The theme that the manuscript deals with could be interesting for the reader and presents innovative aspects, especially with regards to 3D bioprinting.

I encourage the publication of the manuscript as it is.

There are only minor typographical errors in the manuscript

Originality/Novelty: The manuscript proposes, through a discursive review of the literature, an original point of view especially on the theme of the application of Adipose-derived stem cells to 3D bioprinting.

Significance: The discussion of the topic is conducted with scientific consistency and rigor through a discursive review of the literature.

Quality of Presentation: The manuscript is written in an appropriate way and the reference articles are cited correctly and appropriately.

Scientific Soundness: Being a discursive review of the literature, the manuscript cannot be judged according to the parameters that are used in the case of an experimental scientific study. Despite this, the logical thread used for the discussion of the topic appears robust since the results of the scientific articles have been reported correctly and in logical sequence.

Interest to the Readers: In my opinion, the conclusions are very interesting for the readership of the Journal. in fact, in my humble opinion, the manuscript identifies an important and innovative direction for scientific research, namely the use of Adipose-derived stem cells in association with 3D printers for the construction of biological and implantable scaffolds.

Overall Merit: There is an overall benefit to publishing this work.

English Level: The English language is appropriate and clearly understandable.  

Reviewer 3 Report

line 27

should read ASCs are also--missing the word "are"

might be nice to add a sentence or two telling the reader why having telomerase activity is beneficial

line 39 --enable?/allow needs to be fixed

Line 79--1st time "AT" used it should be spelled out -consistency

Line 153- spell out "SC"

Line 235 - I feel that the word "biocompatible" be removed its redundant with the rest of sentence saying "which proves its biocompatibilty"

line 330 - the sentence starting with Thus, they are ...is very awkward and needs to be reworded.

line 335- random "e" after mimicked

line 336 - change "designed to" to read "designed for"

Figures:

   only figure 2 is cited as being adapted from another source-are the other figures from the authors and not previously published? why is the full citation in figure 2 in the legend can't it be the same as an in text number??

Author Response

First, thank you very much for reviewing and evaluating our manuscript.

 Specific Comments

 line 27 should read ASCs are also--missing the word "are"

Thanks for the notes. We did the changes as requested.

might be nice to add a sentence or two telling the reader why having telomerase activity is beneficial

Well pointed. We complete the sentence. “These adipose-derived stem cells (ASCs) are also capable of proliferating at higher rates and show telomerase activity, which acts by maintaining the DNA stability during cell divisions.”

 line 39 --enable?/allow needs to be fixed

The revised version was corrected accordingly.

 Line 79--1st time "AT" used it should be spelled out -consistency

Was cited in line 68: “The MSCs from adipose tissue (AT) are more heterogeneous…”

Line 153- spell out "SC"

We did the changes as requested.

Line 235 - I feel that the word "biocompatible" be removed its redundant with the rest of sentence saying "which proves its biocompatibilty"

Thanks for revision. We removed the redundant word.

line 330 - the sentence starting with Thus, they are ...is very awkward and needs to be reworded.

This sentence has been removed and incorporated into the previous paragraph.

“Changes in the culture settings of ASCs, despite the phenotypic of cells after isolation, can shift the cell profile to several subpopulations, leading to considerable differences in the proliferation rate and differentiation of cells available for therapeutic procedures, opening multiple novel avenues for tissue engineering and regenerative medicine strategies, especially in precision medicine.”

line 335- random "e" after mimicked

We removed the word.

line 336 - change "designed to" to read "designed for"

We did the changes as requested.

Figures:

   only figure 2 is cited as being adapted from another source-are the other figures from the authors and not previously published? why is the full citation in figure 2 in the legend can't it be the same as an in text number??

The other figures are copyright figures and are not adapted.

Reviewer 4 Report

As a review, the authors' personal contribution in novel investigation direction is necessary, which is completely lacking in the current manuscript. 

It is hard to catch the thought of the authors. Particularly, it is hard to track how the sections are arranged and organized. Multiple information is repeated again and again in several different sections, while none of them make a clear and/or complete illustration. 

The link among ACS, biomaterials, and tissue engineering is not established in this manuscript. 

A native English speaker is absolutely necessary to make this manuscript more reader-friendly. 

Some special points which are not exclusive: 

  1. There are two kinds of definitions of MSCs are provided. 
  2. Liposuction is still a dangerous approach, which at least needs to be discussed. 
  3. About the heterogeneous issue, authors cannot use 'heterogeneous' to explain 'heterogeneous'. 
  4. If the principal phenotypic features of ASCs can be altered by a lot of reasons, how do the principal phenotypic features defined?
  5. It is vague that how WAT influences the numbers of ASCs and their microenvironment. The statement made by the authors seems like 'something can change something in some way'. 
  6. All MSCs are somehow considered immune privileged. What are the pros and cons of ASCs?

The authors should get some hints from the above points about how to revise the manuscript, while there are no points for a reviewer to function as a corresponding author. 

Generally, it may be enough for an undergrad student to present at a lab meeting as his/her literature reading report, but hardly fit the criteria as a review article. In-depth and in-detail interpretation of the currently available reports is warranted for publication. 

Author Response

First, thank you very much for reviewing and evaluating our manuscript.

General Comments

As a review, the authors' personal contribution in novel investigation direction is necessary, which is completely lacking in the current manuscript.

It is hard to catch the thought of the authors. Particularly, it is hard to track how the sections are arranged and organized. Multiple information is repeated again and again in several different sections, while none of them make a clear and/or complete illustration.

The link among ACS, biomaterials, and tissue engineering is not established in this manuscript.

A native English speaker is absolutely necessary to make this manuscript more reader-friendly.

We are attaching the letter from the native English-language reviewer of the submitted manuscript (1st version)

Some special points which are not exclusive:

  1. There are two kinds of definitions of MSCs are provided.

Thanks for the comment. We will keep the sentence as set: The defining characteristics of Mesenchymal stem cells (MSCs).  We only use the term MSCs for Mesenchymal Stem Cell, and the abbreviation was explained first. We not cited Mesenchymal Stromal Cell or Medical Signaling Cell (Caplan 2017).

  1. Liposuction is still a dangerous approach, which at least needs to be discussed.

Liposuction, as all other invasive approaches and surgery, may have impacts on the patient's life. Further clinical investigations and analysis about liposuction still need to be done, but this is not the scope of this article.

  1. About the heterogeneous issue, authors cannot use 'heterogeneous' to explain 'heterogeneous'.

For explained ASC heterogeneous, we highlighted various points:

“The ASCs are heterogeneous and exhibit various features, such as proliferation capacity, differentiation potential, expression of specific surface immunophenotypes, and a secretome profile. These variations exist due to multiple factors, including the donor age, sex, body mass index, clinical condition, isolation procedure (liposuction or fat excision), in which body site and depth of each adipose depot, the sample was withdrawn.”

  1. If the principal phenotypic features of ASCs can be altered by a lot of reasons, how do the principal phenotypic features defined?

ASCs, as a type of MSCs, should relies on these genotypic features:

It was established by the International Society for Cellular Therapy (ISCT) that  the positive expression of surface markers CD13, CD29, CD44, CD73, CD90 and CD105, and negative or low expression of HLA-DR, characterizes MSCs[8,9]. The MSCs from adipose tissue (AT) are more heterogeneous [10], exhibit immunomodulatory properties, and differentiation ability similar to that of bone marrow-derived MSCs (BM-MSCs).

Considering that phenotype depends partially on the genotype, ASCs must present these standard features that all mesenchymal stem cells need to present. In addition, we also highlighted that ASCs show some standard features:

To prevent the incorrect use of different terminology, the International Fat Applied Technology Society adopted the term “adipose-derived stem cells” (ASCs) to identify the isolated, plastic-adherent, multipotent cell population obtained from this site.

  1. It is vague that how WAT influences the numbers of ASCs and their microenvironment. The statement made by the authors seems like 'something can change something in some way'.

When we talk about cell therapy, the donor is the main point. Once this is the cell source, several characteristics can be different between different donors. So, in relation to WAT, it will depend on the factors such age and body mass, which directly influences the number of ASC that can be isolated.

  1. All MSCs are somehow considered immune privileged. What are the pros and cons of ASCs?

All sources of MSCs have pros and cons. Regarding ASCs, the pros for their use are mainly the high number of cells that can be isolated from their source, easy to obtain, their plasticity, and the possibility to give a new use to a tissue that would be discarted (the adipose tissue removed by liposuction). All this have been exposed in the manuscript using other words. Cons are the factors related to the donor, which are something that exist in all types of MSCs sources.

The authors should get some hints from the above points about how to revise the manuscript, while there are no points for a reviewer to function as a corresponding author.

Generally, it may be enough for an undergrad student to present at a lab meeting as his/her literature reading report, but hardly fit the criteria as a review article. In-depth and in-detail interpretation of the currently available reports is warranted for publication.

We appreciate the reviewer’s point of view about this manuscript; therefore, we must point out that all aspects were clarified or solved in the revised manuscript.

Round 2

Reviewer 4 Report

The authors have made some improvements to the manuscript. However, the essential part of a review article, in-depth thinking is still lacking. The responses to the previous comments and the interpretation of the currently available data are superficial. It is not fit the criteria of a review article yet.